# Caterpillar Responses to Gustatory Stimuli in Potato Tuber Moths: Electrophysiological and Behavioral Insights

**DOI:** 10.3390/life13112174

**Published:** 2023-11-07

**Authors:** Ni Mu, Jia-Cai Tang, Jing Zhao, Qi-Chun Fu, Yan-Fen Ma, Rui Tang, Wen-Xia Dong

**Affiliations:** 1State Key Laboratory for Conservation and Utilization of Bio-Resources in Yunnan, College of Plant Protection, Yunnan Agricultural University, Kunming 650201, China; mn199902032023@163.com (N.M.); tjc14769635501@163.com (J.-C.T.); sukbergrise@126.com (J.Z.); 2Plant Protection and Quarantine Station of Daguan County in Yunnan Province, Daguan 657400, China; dgzb201002@126.com; 3Department of Agronomy and Biological Science, Dehong Teacher’s College, Mangshi 678400, China; mayanfen2005@126.com; 4Guangdong Key Laboratory of Animal Conservation and Resource Utilization—Guangdong Public Laboratory of Wild Animal Conservation and Utilization, Institute of Zoology, Guangdong Academy of Sciences, Guangzhou 510260, China

**Keywords:** potato tuber moth, single sensillum recording, gustation, feeding

## Abstract

This research investigates how fourth-instar larvae of the potato tuber moth, *Phthorimaea operculella*, respond to plant secondary metabolites (sucrose, glucose, nicotine, and tannic acid) both in terms of gustatory electrophysiology and feeding behavior. The objective is to establish a theoretical foundation for employing plant-derived compounds in potato tuber moth control. We employed single-sensillum recording techniques and dual-choice leaf disk assays to assess the gustatory electrophysiological responses and feeding preferences of these larvae towards the mentioned compounds. Sensory neurons responsive to sucrose, glucose, nicotine, and tannic acid were identified in the larvae’s medial and lateral sensilla styloconica. Neuronal activity was influenced by stimulus type and concentration. Notably, the two types of sensilla styloconica displayed distinct response patterns for sucrose and glucose while they had similar firing patterns towards nicotine and tannic acid. Sucrose and glucose significantly promoted larval feeding, while nicotine and tannic acid had significant inhibitory effects. These findings demonstrate that the medial and lateral sensilla styloconica house sensory neurons sensitive to both feeding stimulants and inhibitors, albeit with differing response profiles and sensitivities. This study suggests that sucrose and glucose are promising candidates for feeding stimulants, while nicotine and tannic acid show potential as effective feeding inhibitors of *P. operculella* larvae.

## 1. Introduction

The potato tuber moth (*Phthorimaea operculella*), a notable Lepidopteran pest, has a global impact on potato crops. Also known as the tobacco splitworm, this adaptable insect thrives in over 90 tropical and subtropical countries across the Americas, Africa, Australia, and Asia [1,2]. It is known for its adaptability, prolific reproduction, and lack of diapause, making it a potato pest that receives considerable attention [3]. In China, its earliest documented infestation dates back to 1937 when it damaged tobacco in Guangxi’s Liuzhou region. Since then, it has spread widely across potato-growing regions, particularly affecting the Yunnan–Guizhou–Sichuan region [4]. This oligophagous insect primarily feeds on solanaceous crops such as potatoes, tobacco, tomatoes, eggplants, and peppers, with potatoes and tobacco being its most vulnerable hosts [5]. The larvae of the species cause damage by mining leaf tissues and tunneling into tubers, leading to surface deformities and rot. Without refrigeration, crop losses can reach 50–100% [6]. Presently, chemical pesticides are the primary means of control, but their effectiveness is limited due to the pest’s leaf-mining and tunneling behavior. The overuse of chemicals has also led to resistance and environmental concerns. Consequently, there is growing importance in understanding the potato tuber moth’s feeding mechanisms and implementing integrated pest management strategies to combat it [2].

Herbivorous insects and their host plants have coevolved over extended periods, forging intricate relationships. Upon contact with plants, insects utilize their gustatory receptors to perceive nutritional components and secondary metabolites, allowing them to make “accept” or “reject” choices. By identifying substances that induce or deter insect feeding, it becomes possible to develop insect attractants or repellents through their combinations, thus opening up new avenues for beneficial insect rearing and pest control [7,8]. The vinegar fly, *Drosophila melanogaster*, is considered an ideal model organism for studying insect gustation. Research has shown that L-type sensilla on the labellum of *D. melanogaster* exhibit electrophysiological responses to sugar, water, and salt [9]. S-type sensilla are sensitive to the bitter substance cucurbitacin B [10]. Furthermore, studies have revealed that L4-, L6-, and L8-type gustatory receptors on the labellum of *Drosophila melanogaster* respond electrophysiologically to carboxylic acids such as glycolic acid, citric acid, and lactic acid [11]. In *D. suzukii*, I-type gustatory sensilla on the labellum exhibit electrophysiological responses to glucose [12]. Insect attractants, often referred to as feeding stimulants, encompass substances that stimulate insect feeding, such as sugars, inositol, and certain host-specific secondary metabolites. Conversely, plant-derived feeding deterrents inhibit and deter insect feeding. Most active compounds within these deterrents are the secondary metabolites of plants. Notably, alkaloids, phenolics, and terpenoids are the primary secondary metabolites recognized for their feeding-deterrent effects on insects [13]. Sugars, as a fundamental energy source for organisms, significantly influence food palatability and serve as widespread feeding stimulants [14]. Nicotine is a pyridine alkaloid primarily produced by Solanaceae plants, particularly tobacco. It plays a pivotal role as a feeding deterrent in various insect species [15,16]. Tannic acid, a type of polymeric phenolic compound, imparts an astringent taste and is prevalent in various plant organs and structures [17].

Most Lepidopteran larvae possess three to five functionally distinct sensory neurons, including mechanical and gustatory receptor neurons (GRNs), within each sensillum on their mouthparts. Various types of GRNs can perceive different chemical signals, guiding insects in making feeding choices [18]. Researchers have utilized gustatory single-sensillum electrophysiology to study how certain Lepidopteran larvae respond to different plant secondary metabolites. For instance, *Spodoptera littoralis* and *Ostrinia nubilalis* exhibit responses in their medial and lateral sensilla styloconica to certain steroids [19]. *S. frugiperda* shows electrophysiological responses in its medial and lateral sensilla styloconica to substances such as matrine, azadirachtin, and rotenone. Similarly, *S. litura* displays strong electrophysiological responses in its medial and lateral sensilla styloconica to saponins [20,21]. In *Helicoverpa armigera* and *H. assulta*, the lateral sensilla styloconica house sensitive neurons responding to sugars, including sucrose, fructose, glucose, and proline, albeit with varying intensities [22]. *Agrotis ipsilon* displays electrophysiological responses in both its medial and lateral sensilla styloconica to compounds such as chlorogenic acid, fumaropimarie acid, and strychnine [23]. Additionally, *Plutella xylostella* exhibits electrophysiological responses in its medial sensilla styloconica to compounds such as sinigrin, brassinolide, and 24-epibrassinolide [24]. Meanwhile, *Pieris rapae* demonstrates responses in its lateral sensilla styloconica to compounds such as sinigrin and four other glucosinolates, with the medial sensilla styloconica specifically responsive to glucobrassicin [25].

However, there are currently no reported gustatory single-sensillum electrophysiology data for potato tuber moth larvae. In this study, we employ single-sensillum recording techniques to investigate two pairs of sensilla on potato tuber moth larvae mouthparts, aiming to understand their gustatory mechanisms towards sucrose, glucose, nicotine, and tannic acid. Additionally, we employ dual-choice leaf disk assays to assess how these substances affect larval feeding behavior, contributing to the theoretical and practical knowledge base for potato tuber moth control.

## 2. Materials and Methods

### 2.1. Insects and Chemicals

*P. operculella* larvae were collected from Dehong, Yunnan (E 97°58′, N 24°47′), and reared using potato tubers as their food source. After emergence, the adult moths were fed a 10% honey solution. Indoor rearing conditions were maintained at a temperature of (26 ± 2) °C, with a light–dark cycle of 14 h of light and 10 h of darkness, and a relative humidity range of 50–70%. Chemicals were purchased accordingly and prepared for later use (Table 1). In the single-sensillum recording electrophysiology experiments, a 2 mmol/L NaCl solution was prepared as the buffer, to which sucrose, glucose, nicotine, and tannic acid were added to create stimulating solutions at concentrations of 0.1 mmol/L, 1 mmol/L, 2 mmol/L, 5 mmol/L, and 10 mmol/L, respectively. For the behavioral assays, a 2 mmol/L NaCl solution was prepared as the buffer, and sucrose, glucose, nicotine, and tannic acid were added to create solutions at concentrations of 10 mmol/L, 30 mmol/L, and 50 mmol/L, respectively.

### 2.2. Single-Sensillum Recording

Electrophysiological measurements of larval gustatory sensors in response to four chemical substances were conducted using single-sensillum recording [26]. Fourth-instar larvae were subjected to 1–2 h of fasting prior to the tests. During this time, the larvae were in a non-fasted yet non-satiated state to ensure consistent conditions among different individuals. Tested larvae had a portion of their body removed after the second thoracic segment. A tungsten wire electrode (Rush Metal Co. Ltd., Shang-hai, China) was sharpened and inserted directly into the larval head as a reference electrode, extending the larval sensilla styloconica outward. Glass capillaries (TW100-4, World Precision Instruments, Friedberg, Germany) were stretched into two equal segments using a microelectrode puller (Model P-2000, Sutter Instrument, Novato, CA, USA). These capillaries served as the recording electrodes, filled with stimulating solutions. The tungsten wire reference electrode was sharpened to a tip with a diameter of approximately 5–10 μm. The glass capillary recording electrode was tapered to a tip diameter of about 10 μm using a micropipette grinder (EG-401, Narishige, Tokyo, Japan). They were connected to a signal amplifier (Syntech, Buchenbach, Germany) and a taste probe (Taste PROBE DTN-02, Syntech, Buchenbach, Germany). The other end was placed over the terminal end of the sensilla styloconica with the assistance of a micromanipulator (Ruiqi Life Science Instrument Co. Ltd., Jiang-Su, China) (Appendix A). The positions and morphology of the medial and lateral sensilla styloconica on the mandibular maxillae of fourth-instar potato tuber moth larvae are shown in Appendix A. Circuit activation was achieved via light pedal touch. AutoSpike 32 software (Syntech, Buchenbach, Germany) was used to record response potentials. Stimulation lasted for 5 s, with each concentration tested on each larva repeated three times and an interval of at least 60 s between two stimulations. Concentrations of each substance were applied in increasing order, with at least seven larvae tested for each substance. The recordings encompassed all spikes within the first second following stimulation (total response) and the number of responses for each firing pattern. Each active neuron’s response frequency was identified based on the principle that a certain amplitude of firing spike corresponds to one active neuron [27].

### 2.3. Behavioral Assay

The larval feeding preference behavior was determined using a dual-choice leaf disk assay [28]. Fourth-instar larvae that had been starved for 4–6 h were selected for the experiment. For the test, 9 cm diameter Petri dishes were prepared, with filter paper moistened with distilled water placed inside. Healthy and uniformly grown tobacco leaves (tobacco variety: K326; number of leaves per plant: 12 ± 3) were chosen, cleaned, and air-dried. Using a 6 mm diameter hole punch, the leaves were formed into leaf disks. These leaf disks were divided into two groups: the control group (A) and treatment group (B). The leaf disks in the treatment group were soaked in a treatment solution, while those in the control group were soaked in a control solution of only 2 mmoL/L NaCl for 40 min. Afterward, they were slightly air-dried. The leaf disks from both groups were placed along the edge of the filter paper at equal distances in an ABABABAB sequence. The positions of each group of leaf disks were marked. Finally, starved larvae were placed in the center of the filter paper, with one larva per Petri dish (Appendix A). As the larvae began feeding, their feeding behavior was intermittently observed. For each larva, when it consumed approximately half of the area of either the treatment or control leaf disk first, its feeding was stopped, and the larva was removed from the Petri dish. The feeding areas of the treatment and control leaf disks were measured using ImageJ software (Appendix A). Each naïve larva was used only once, and at least 30 larvae were tested for each chemical.

The feeding preference index for larvae regarding treatment and control leaf disks was calculated using the following formulae:Control Feeding Preference Index (%) = (Area of control group leaf disks consumed in the dish/Total area of all leaf disks consumed in the dish) × 100;
Treatment Feeding Preference Index (%) = (Area of treatment group leaf disks consumed in the dish/Total area of all leaf disks consumed in the dish) × 100

### 2.4. Data Processing

Statistical analysis and data visualization were conducted using Microsoft Excel (RRID:SCR_016137), SPSS 26 (RRID:SCR_002865), and ImageJ (RRID:SCR_003070). Descriptive statistics were employed to verify whether or not the data followed a normal distribution. Subsequently, a one-way analysis of variance (ANOVA) was performed, followed by Tukey HSD multiple comparison tests, to examine the electrophysiological response frequencies of the two types of sensilla to various tested compounds and the larval feeding preference index. A significance level of 0.05 was selected to determine significant differences, while a threshold of 0.01 was applied to indicate highly significant differences. Graphs and plots were generated using Origin 2021 (RRID:SCR_002815).

## 3. Results

### 3.1. Overall Comparison of Larval Sensilla

The single-sensillum recording results indicated that all four tested compounds could induce relatively strong electrophysiological responses in both medial and lateral sensilla. However, the intensity of responses varied with different substances (Figure 1).

At a concentration of 0.1 mmol/L, the medial sensilla styloconica exhibited significantly higher responses to sucrose compared to the other three compounds. At a concentration of 1 mmol/L, the medial sensilla styloconica displayed significantly higher responses to both sucrose and glucose compared to tannic acid and nicotine. At concentrations of 2 mmol/L and 10 mmol/L, the medial sensilla styloconica showed significantly higher responses to nicotine compared to tannic acid, with no significant difference in responses between sucrose and glucose. At a concentration of 5 mmol/L, there was no significant difference in responses to the four tested compounds by the medial sensilla styloconica (Figure 2A).

At a concentration of 0.1 mmol/L, the lateral sensilla styloconica exhibited significantly higher responses to sucrose compared to those for the other three compounds. At a concentration of 1 mmol/L, the lateral sensilla styloconica displayed significantly higher responses to glucose compared to those for nicotine, with no significant differences in responses between sucrose and tannic acid. At concentrations of 2 mmol/L and 5 mmol/L, the lateral sensilla styloconica showed significantly higher responses to both sucrose and glucose compared to those for tannic acid and nicotine, but the differences between them were not significant. At a concentration of 10 mmol/L, the lateral sensilla styloconica exhibited the highest response to glucose, which was significantly higher than its response to the other three compounds (Figure 2B).

### 3.2. Responses of Medial Sensilla Styloconica

Each set of raw response data obtained through the Autospike 32 software was analyzed to determine the nature and quantity of taste neurons involved in the responses [21,22]. The “amplitude” classification feature within the software was utilized to differentiate spikes based on amplitude levels. Following the principle that each amplitude corresponds to a specific neuron, it was determined that there were three taste neurons activated within the medial sensilla styloconica for each stimulation solution, albeit with varying properties and response frequencies.

Sucrose and glucose solutions activated neurons I, II, and IV within the medial sensilla styloconica. Neuron II was identified as the D-Glucose and sucrose-sensitive neuron (Figure 3A,B). In contrast, nicotine and tannic acid solutions activated neurons I, III, and IV, with neuron III being identified as the inhibitor-sensitive neuron (Figure 3C,D). Neurons I and IV were found to be responsive to water and low-concentration salt.

In later dosage response tests, the total spikes of the medial sensilla styloconica to the four stimulants were similar. They all exhibited an increase in total response with increasing of concentrations, reaching a peak at a concentration of 1 mmol/L. Subsequently, as the concentration of the stimulant substance increased, the total response gradually decreased (Figure 3A–D). However, there were differences in the response intensities induced by different stimulant substances.

For the sucrose solution, the total response of all taste receptor neurons within the medial sensilla styloconica induced by 0.1 mmol/L, 1 mmol/L, and 2 mmol/L sucrose solutions was significantly higher than the response frequency induced by the control 2 mmol/L NaCl solution. However, there were no significant differences in the total responses induced by 5 mmol/L and 10 mmol/L sucrose solutions compared to those induced by the control solution (Figure 3A). At different concentrations of sucrose solution, the firing intensities of neurons I and IV within the medial sensilla styloconica were relatively low (Figure 3A), while the response pattern of neuron II induced by sucrose was similar to the overall response pattern of the medial sensilla styloconica to sucrose (Figure 3A).

For the glucose solution, the total response of all taste receptor neurons within the medial sensilla styloconica induced by 0.1 mmol/L, 1 mmol/L, 2 mmol/L, and 5 mmol/L glucose solutions was significantly higher than the response induced by the control solution. However, there were no significant differences in the total response frequencies induced by 10 mmol/L glucose solution compared to those induced by the control solution (Figure 3B). At different concentrations of glucose solution, the firing intensities of neurons I and IV within the sensilla were relatively low (Figure 3B), while the response pattern of neuron II induced by glucose was similar to the overall response pattern of the medial sensilla styloconica to glucose (Figure 3B).

For the nicotine solution, the total responses of all taste receptor neurons within the sensilla induced by 0.1 mmol/L, 1 mmol/L, and 2 mmol/L nicotine solutions was significantly higher than the responses induced by the control solution. However, there were no significant differences in the total responses induced by 5 mmol/L and 10 mmol/L nicotine solutions compared to those induced by the control solution (Figure 3C). At different concentrations of nicotine solution, the response intensities of neurons I and IV within the sensilla were relatively low (Figure 3C), while the response pattern of neuron III induced by nicotine was similar to the overall response pattern of the medial sensilla to nicotine (Figure 3C).

For the tannic acid solution, the total responses of all taste receptor neurons within the sensilla induced by 1 mmol/L and 2 mmol/L tannic acid solutions was significantly higher than the response frequency induced by the control solution. However, there were no significant differences in the total response frequencies induced by 0.1 mmol/L and 5 mmol/L tannic acid solutions compared to those induced by the control solution, and the total response frequency induced by the 10 mmol/L tannic acid solution was significantly lower than that induced by the control solution (Figure 3D). At different concentrations of tannic acid solution, the response intensities of neurons I and IV within the sensilla were relatively low (Figure 3D), while the response pattern of neuron III induced by tannic acid was similar to the overall response pattern of the medial sensilla to tannic acid (Figure 3D).

### 3.3. Responses of Lateral Sensilla Styloconica

For the lateral sensilla styloconica, sucrose, glucose, nicotine, and tannic acid all activated the responses of three neurons. Sucrose and glucose solutions could activate neurons I, II, and IV (Figure 4A,B), while nicotine and tannic acid solutions activated neurons I, III, and IV within the lateral sensilla styloconica (Figure 4C,D).

The lateral sensilla styloconica of the larvae exhibited dose-dependent response patterns for both sucrose and glucose, with increasing stimulus concentrations leading to progressively stronger responses. In the case of sucrose, concentrations of 0.1 mmol/L, 1 mmol/L, 2 mmol/L, 5 mmol/L, and 10 mmol/L induced significantly higher total response frequencies in the sensilla compared to those induced by the control (Figure 4A). The firing of neurons I and IV within the sensilla was lower at different concentrations of sucrose (Figure 4A), while neuron II exhibited a response pattern similar to the overall response to sucrose, demonstrating a dose-dependent pattern (Figure 4A).

Similarly, for glucose, concentrations of 0.1 mmol/L, 1 mmol/L, 2 mmol/L, 5 mmol/L, and 10 mmol/L induced significantly higher total response frequencies compared to those induced by the control (Figure 4B). The firing of neurons I and IV was lower at different concentrations of glucose (Figure 4B). Neuron II displayed a response pattern similar to the overall response to glucose and also exhibited a dose-dependent pattern (Figure 4B).

For the nicotine solution, concentrations of 0.1 mmol/L, 1 mmol/L, 2 mmol/L, 5 mmol/L, and 10 mmol/L induced significantly higher total response frequencies in the lateral sensilla styloconica compared to those induced by the control solution (Figure 4C). The response strength of neurons I and IV was lower at different concentrations of nicotine (Figure 4C), while neuron III exhibited a response pattern similar to the overall response to nicotine (Figure 4C).

For the tannic acid solution, concentrations of 0.1 mmol/L and 10 mmol/L did not induce significantly different total response frequencies compared to those induced the control solution, while concentrations of 1 mmol/L, 2 mmol/L, and 5 mmol/L induced significantly higher total response frequencies compared to those induced by the control solution (Figure 4D). The response strength of neurons I and IV was lower at different concentrations (Figure 4D), while neuron III exhibited a response pattern similar to the overall response to tannic acid (Figure 4D).

### 3.4. Feeding Preferences of Larvae Regarding Four Plant Metabolites

In the case of sucrose and glucose, as the concentration of the stimulus solution increased, the larvae exhibited an elevated feeding preference for the treated leaf discs. For 10 mmol/L, 30 mmol/L, and 50 mmol/L sucrose-treated leaf discs, the larval feeding preference indexes were 0.4913 ± 0.2788, 0.6311 ± 1.9658, and 0.6587 ± 2.4599, respectively. Similarly, for 10 mmol/L, 30 mmol/L, and 50 mmol/L glucose-treated leaf discs, the larval feeding preference indexes were 0.4794 ± 0.4561, 0.6701 ± 2.1345, and 0.6851 ± 2.3565, respectively. Notably, there was no significant difference in larval feeding preference indexes between 10 mmol/L sucrose or glucose-treated leaf discs and control leaf discs, whereas the indexes for 30 mmol/L and 50 mmol/L sucrose and glucose-treated leaf discs were significantly higher than those for control leaf discs (Figure 5A,B).

As the concentration of plant secondary metabolites nicotine and tannic acid increases, larvae show a gradual decrease in their feeding preference for the treated discs and an increase in their preference for control discs. The feeding selection index of larvae for discs treated with 10 mmol/L, 30 mmol/L, and 50 mmol/L of nicotine or tannic acid is as follows: 0.4772 ± 0.9657, 0.3296 ± 2.0872, and 0.3061 ± 1.4132 for nicotine, and 0.4838 ± 0.3657, 0.3667 ± 0.8177, and 0.3082 ± 1.0561 for tannic acid. Among these, there is no significant difference in the feeding preference index of larvae for 10 mmol/L nicotine or tannic acid-treated discs compared to control discs, while the feeding preference index for 30 mmol/L and 50 mmol/L nicotine or tannic acid-treated discs is significantly lower than that for control discs (Figure 5C,D).

## 4. Discussion

In general, it is believed that the two pairs of sensilla on the mouthparts of lepidopteran larvae, the sensilla styloconica, primarily serve the functions of host discrimination and regulating feeding quantities during their feeding processes [29]. These sensilla typically consist of one gustatory receptor neuron sensitive to sugars and other feeding stimulants and one gustatory receptor neuron sensitive to bitter compounds and other deterrents. The activity of these neurons often plays a crucial role in determining the feeding choices of insect larvae [30].

Different species and closely related species of insects may have variations in the types of gustatory receptors present in their taste sensilla. For example, closely related insects such as the cotton bollworm and the oriental tobacco budworm, both belonging to the family Noctuidae, exhibit differences in their gustatory sensitivity to sucrose [31]. It was shown that the tobacco hornworm exhibits greater sensitivity to sucrose compared to that of the cotton bollworm via its lateral sensilla styloconica. Furthermore, the minimum threshold concentration of sucrose that significantly induces larval feeding preference in cotton bollworm is higher than that for the tobacco hornworm. Similarly, two closely related butterfly species, *Papilio hospiton* and *P. machaon*, display significant differences in the chemical sensitivity of their larval taste sensilla [32]. Larvae from these two species revealed completely opposite sensitivities to sucrose and glucose of the lateral sensilla styloconica.

This study has revealed specific characteristics of the electrophysiological responses of mouthpart sensilla in the larvae of the potato tuber moth to various plant secondary metabolites. We found that both sensilla styloconica contain gustatory receptor neurons sensitive to sucrose, glucose, nicotine, and tannic acid. Furthermore, the overall responses of the lateral sensilla styloconica to these four substances are stronger than those of the medial sensilla styloconica. Additionally, the response patterns to sucrose and glucose differ between the two sensilla styloconica.

Generally, a solution of a particular substance can activate the activity of three neurons within the larval sensilla styloconica. However, the neurons sensitive to the specific substance exhibit varied activity. These findings demonstrate that the gustatory responses of potato tuber moth larvae exhibit both commonalities and specificities when compared to other lepidopteran insects. The spectrum of responses in the gustatory sensilla on the labial and maxillary palps can vary between species, highlighting the need for specific considerations when studying feeding stimulants and deterrents in different insect species.

The potato tuber moth, as a typical oligophagous insect, primarily feeds on plants from the Solanaceae family, which contain nicotine. Research has shown that feeding potato tuber moth larvae with artificial diets containing 0.2% nicotine significantly inhibits larval weight gain and pupal weight, leading to an extended generational period. This inhibitory effect is more pronounced in younger larvae [33]. The findings of this study indicate that potato tuber moth larvae can recognize plant metabolites in their food. As natural tobacco leaves presented traceable level of tested stimulants [34,35], it highly suggested that sucrose and glucose exhibit a significant preference for feeding on leaf discs, while nicotine and tannic acid suppress the feeding behavior of potato tuber moth larvae. These findings are similar to the feeding choices observed in other lepidopteran insects [36,37,38,39,40]. Sugars such as sucrose and glucose are crucial metabolic products of plant photosynthesis and play a vital role in the feeding behavior of Lepidoptera insects. Research has found that in most Lepidoptera larvae, including *H. armigera*, *H. assulta*, *Busseola fusca*, *S. frugiperda*, the sensilla styloconica on their mouthparts exhibit strong electrophysiological responses to sucrose and glucose, resulting in a strong feeding preference [31,41,42]. Therefore, for the majority of Lepidoptera insects, sucrose and glucose serve as potent feeding stimulants. This conclusion aligns with the findings in this study regarding the feeding-stimulating effects of sucrose and glucose on potato tuber moth larvae.

Furthermore, our results collectively suggest that certain concentrations of secondary metabolites can act as feeding deterrents for insects. However, one should notice that host preference in insects is not solely determined by taste but also be influenced by the other factors including nutritional value such as D-glucose vs. L-glucose, and decision making in insects involves a genetic factor and environmental factor [21]. Some studies have indicated that oligophagous insects can use specific secondary metabolites unique to their host plants as token cues for host recognition and acceptance [43]. For example, the secondary metabolite glycoalkaloid in Solanaceae plants attracts tobacco hornworms [44], and cardiac glycosides serve as signature cues that induce *Danaus plexippus* to feed on *Asclepias genus* [45]. In this study, we provided fundamental information for potato tuber moths in the gustatory reception of selected tastants. Future deorphanization investigations of gustatory receptors (GRs) within this species can benefit from our results on peripheral coding patterns for these chemicals. Further explorations using either heterologous expressions or transgenic lines to unveil precise coding mechanisms of GRs and ionotropic receptors (IRs) will need to be addressed in this species [46].

Tannic acid is a polyphenolic secondary metabolite found in plants that imparts astringency when consumed, making it unpalatable. Research has indicated that tannic acid can directly interact with taste receptors, leading to reduced feeding and, consequently, reduced growth and development in animals [47]. Tannic acid has been shown to inhibit the feeding of various insect larvae, including the fall armyworm [48,49,50]. We observed that potato tuber moth larvae’s taste receptors exhibit electrophysiological responses to tannic acid, which inhibits the feeding of potato tuber moth larvae. Tannic acid is known to be an effective inhibitor of various enzyme-catalyzed reactions and can inhibit the activity of several digestive enzymes, thereby affecting larval feeding and growth. It is considered an antinutritional factor [51]. Tannic acid can also inhibit the activity of tyrosinase, leading to incomplete sclerotization and the darkening of the insect cuticle [52]. However, in this study, no such cuticle changes were observed in potato tuber moth larvae. Therefore, it is suggested that tannic acid may not affect the activity of tyrosinase in potato tuber moth larvae, or it could be due to the relatively short feeding duration in this study, resulting in no visible changes in the larval cuticle. Additionally, exposure to tannic acid has been shown to enhance the activities of detoxifying enzymes such as glutathione S-transferase, cytochrome P450 enzymes, and carboxylesterases in the larvae of various insects [48,53,54,55]. However, it remains unclear whether or not exposure to tannic acid induces changes in the activity of detoxification enzymes in potato tuber moth larvae.

Nicotine has neurotoxic effects on most herbivorous insects, primarily acting on their acetylcholine receptors, which disrupts normal nerve impulse transmission, leading to neural excitation and eventual disruption [56]. Studies have found that nicotine significantly inhibits the activity of acetylcholinesterase in the head tissues of fall armyworm larvae, resulting in intoxication and death [57]. In this work, it was observed that potato tuber moth larvae exhibited partial mortality after feeding on tobacco leaves treated with high concentrations of nicotine while no dead larva was observed for tannic acid treatment (Appendix A). This occurrence may be attributed to the neurotoxic effect of nicotine on potato tuber moth larvae. Besides its neurotoxicity, nicotine also possesses antifeedant properties and inhibits protein digestion in insects [58]. Moreover, the volatilization of nicotine has been documented [59,60], suggesting the potential for olfactory reception in potato tuber moth larvae. Nevertheless, our experiments have meticulously examined specific feeding locations, significantly minimizing pre-contact selections influenced by olfactory perception. Research has shown that nicotine significantly affects the gut microbial cell structure and function of *Dendrolimus superans* and inhibits gut protease activity [57]. In this study, it was found that low concentrations of nicotine with electrophysiological responses did not affect the feeding behavior of potato tuber moth larvae. The electrophysiological responses of larval gustatory sensilla do not necessarily align with their feeding choices. Feeding preferences depend on factors beyond stimulus concentration, including nutritional value, genetic factors, and environmental variables. This lack of impact may be due to the significant increase in the activity of detoxifying enzymes such as carboxylesterases and glutathione S-transferases in the larvae’s bodies after nicotine ingestion, enhancing their ability to detoxify toxic substances [61].

In terms of practical applications, adding glucose and sucrose to insecticides can enhance their palatability to potato tuber moth larvae, making them more receptive and thereby increasing ingestion, which augments the lethal effect. High concentrations of tannic acid and nicotine have inhibitory effects on potato tuber moth larvae’s feeding behavior by disturbing the recognition of their taste neurons, rendering them unable to recognize food. Including these compounds in insecticide formulation can better halt the damage to protected plants and stored products. This approach proves more effective in safeguarding crops than conventional insecticides alone.

## 5. Conclusions

This study represents the first investigation into the taste physiology and feeding behavior of potato tuber moth larvae concerning typical feeding stimulants and inhibitors. It demonstrates that larval feeding choices are greatly influenced by insect taste perception and highlights the effectiveness of plant secondary metabolites in inhibiting insect feeding. These findings contribute to our understanding of the foundational principles behind insect feeding choices and lay the theoretical groundwork for utilizing plant secondary metabolites in the control of potato tuber moth. Given the diversity of secondary metabolites found in different host plants, future research can continue to screen potential feeding stimulants and inhibitors that may be effective against potato tuber moth. Further research is also needed to elucidate the mechanisms behind the larval taste perception of these substances. Building upon this foundation, studying the synergistic effects and mechanisms of different combinations of feeding stimulants or inhibitors could be a promising research direction for identifying active compounds.

## Figures and Tables

**Figure 1 life-13-02174-f001:**
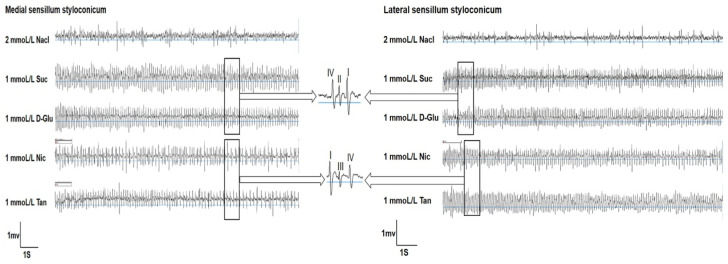
Representative traces of single-sensillum electrophysiological activity of the gustatory receptor neurons in both types of sensilla styloconica of *P. operculella* larvae in response to different phagostimulants and deterrents. Suc: sucrose; D-Glu: D-Glucose; Nic: nicotine; Tan: tannic acid.

**Figure 2 life-13-02174-f002:**
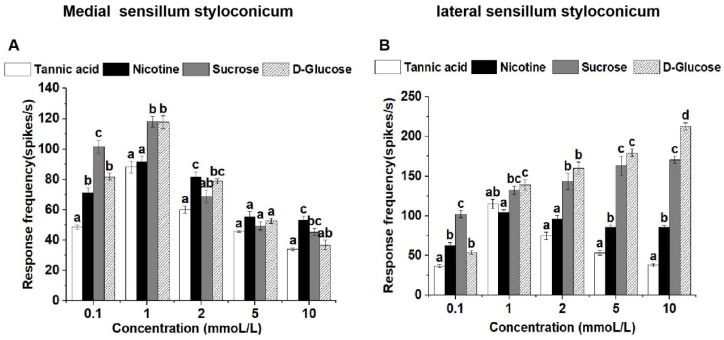
Comparison of single-sensillum electrophysiological responses in medial sensilla styloconica (**A**) and lateral sensilla styloconica (**B**) on *P. operculella* larvae mouthparts to four stimulants. Lower case letters indicate significant differences among dosages (ANOVA and Tukey HSD comparison, medial sensilla styloconica, 0.1 mmoL/L: F_3, 29_ = 43.587, *p* = 0.039; 1 mmoL/L: F_3, 30_ = 17.858, *p* = 0.961; 2 mmoL/L: F_3, 16_ = 10.257, *p* = 0.046; 5 mmoL/L: F_3, 16_ = 2.727, *p* = 0.058; 10 mmoL/L: F_3, 26_ = 10.537, *p* = 0.301. lateral sensilla styloconica, 0.1 mmoL/L: F_3, 28_ = 53.650, *p* = 0.013; 1 mmoL/L: F_3, 26_ = 9.535, *p* = 0.367; 2 mmoL/L: F_3, 18_ = 30.411, *p* = 0.147; 5 mmoL/L: F_3, 19_ = 75.730, *p* = 0.000; 10 mmoL/L: F_3, 26_ = 363.497, *p* = 0.121).

**Figure 3 life-13-02174-f003:**
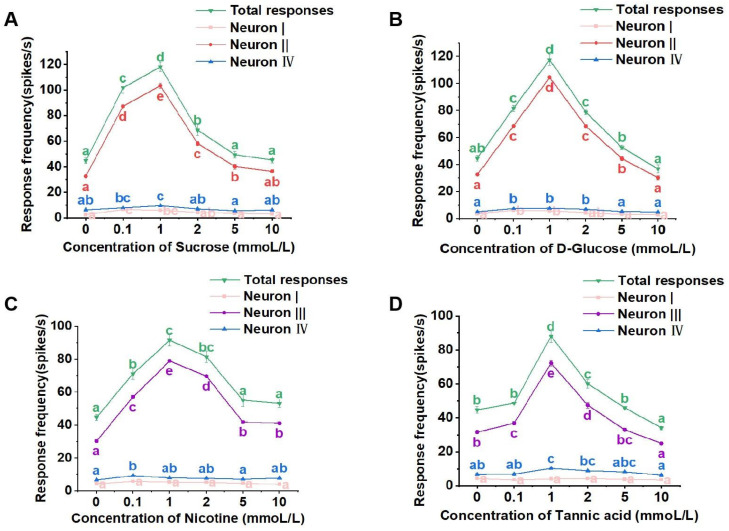
Electrophysiological response of the medial sensillum styloconicum of the 4th instar larvae of *P. operculella* to sucrose (**A**), D-glucose (**B**), nicotine (**C**), and tannic acid (**D**), respectively. Lower-case letters indicate significant differences among dosages (ANOVA and Tukey HSD comparison, Suc: total responses, F_5, 44_ = 102.293, *p* = 0.243; neuron I, F_5, 36_ = 12.027, *p* = 0.724; neuron II, F_5, 36_ = 399.055, *p* = 0.481; neuron IV, F_5, 36_ = 8.540, *p* = 0.423; D-Glu: total responses, F_5, 33_ = 122.234, *p* = 0.048; neuron I, F_5, 36_ = 8.918, *p* = 0.582; neuron II, F_5, 36_ = 605.673, *p* = 0.486; neuron IV, F_5, 36_ = 11.477, *p* = 0.559; Nic: total responses, F_5, 44_ = 37.630, *p* = 0.631; neuron I, F_5, 36_ = 1.981, *p* = 0.180; neuron III, F_5, 36_ = 503.633, *p* = 0.610; neuron IV, F_5, 36_ = 4.061, *p* = 0.876; Tan: total responses, F_5, 32_ = 65.802, *p* = 0.006; neuron I, F_5, 36_ = 0.594, *p* = 0.190; neuron III, F_5, 36_ = 201.506, *p* = 0.217; neuron IV, F_5, 36_ = 7.150, *p* = 0.704).

**Figure 4 life-13-02174-f004:**
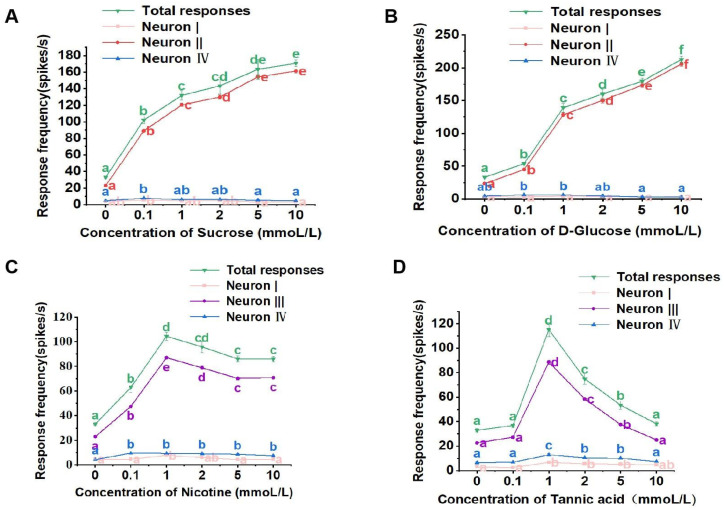
Electrophysiological response of the lateral sensillum styloconicum of the 4th instar larvae of *P. operculella* to sucrose (**A**), D-glucose (**B**), nicotine (**C**), and tannic acid (**D**), respectively. Lower-case letters indicate significant differences among dosages (ANOVA and Tukey HSD comparison, Suc: total responses, F_5, 46_ = 81.728, *p* = 0.000; neuron I, F_5, 36_ = 4.814, *p* = 0.747; neuron II, F_5, 36_ = 570.030, *p* = 0.015; neuron IV, F_5, 36_ = 6.549, *p* = 0.637; D-Glu: total responses, F_5, 31_ = 311.233, *p* = 0.151; neuron I, F_5, 36_ = 1.044, *p* = 0.288; neuron II, F_5, 36_ = 641.535, *p* = 0.001; neuron IV, F_5, 36_ = 7.556, *p* = 0.107; Nic: total responses, F_5, 44_ = 78.621, *p* = 0.050; neuron I, F_5, 36_ = 6.151, *p* = 0.066; neuron III, F_5, 36_ = 52.423, *p* = 0.114; neuron IV, F_5, 36_ = 15.410, *p* = 0.508; Tan: total responses, F_5, 32_ = 100.584, *p* = 0.000; neuron I, F_5, 36_ = 9.349, *p* = 0.193; neuron III, F_5, 36_ = 500.090, *p* = 0.090; neuron IV, F_5, 36_ = 22.415, *p* = 0.476.).

**Figure 5 life-13-02174-f005:**
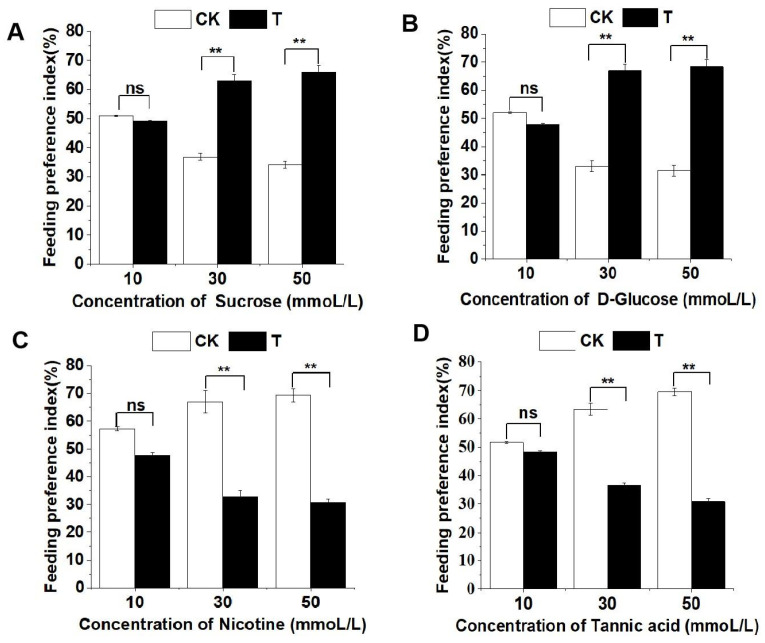
Feeding preference of the 4th instar larvae of *P. operculella* for sucrose (**A**), D-glucose (**B**), nicotine (**C**), and tannic acid (**D**), respectively. Asterisks indicate significant differences between treatments in respective dosages (*t* test. ‘ns’ indicates no significant difference observed, and ‘**’ indicates significant difference observed with *p* < 0.01. Suc: 10 mmoL/L, *t* = 0.0299, *p* = 0.8626; 30 mmoL/L, *t* = 6.8764, *p* = 0.0087; 50 mmoL/L, *t* = 10.0725, *p* = 0.0015; D-Glu: 10 mmoL/L, *t* = 0.1694, *p* = 0.6806; 30 mmoL/L, *t* = 11.5757, *p* = 0.0007; 50 mmoL/L, *t* = 13.7200, *p* = 0.0002; Nic: 10 mmoL/L, *t* = 0.8717, *p* = 0.3504; 30 mmoL/L, *t* = 11.6145, *p* = 0.0006; 50 mmoL/L, *t* = 15.0380, *p* = 0.0001; Tan: 10 mmoL/L, *t* = 0.1043, *p* = 0.7466; 30 mmoL/L, *t* = 7.1098, *p* = 0.0076; 50 mmoL/L, *t* = 14.9858, *p* = 0.0001).

**Table 1 life-13-02174-t001:** Purity and source of standard chemical compounds.

Compounds	Purity (%)	CAS	Source
Sucrose	Analytic pure	57-50-1	Wind Boat chemical reagent Technology Co. Ltd., Tian-jin, China
D-Glucose	Analytic pure	14431-43-7	Xilong Scientific Co. Ltd., Shan-tou, China
Nicotine	≥99.0	54-11-5	Neobioscience, Shen-zhen, China
Tannic acid	99.0	1401-55-4	J&K Scientific, San Jose, CA, USA
Sodium chloride	Analytic pure	7647-14-5	Xilong Scientific Co. Ltd., Shan-tou, China

## Data Availability

All data described in the current work are available through access to the text or Appendix A.

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
