# Peer review of "Caterpillar Responses to Gustatory Stimuli in Potato Tuber Moths: Electrophysiological and Behavioral Insights"

_life, 2023, doi:10.3390/life13112174_

Round 1

Reviewer 1 Report

Comments and Suggestions for Authors

The authors explored key questions on gustatory reception and behavioral valences of an important potato pest Phthorimaea operculella larvae towards selected plant metabolites. They first assessed the gustatory electrophysiological responses using single sensillum recording, followed by studying feeding preferences by dual-choice leaf disk assays. Below comments which may be helpful to the authors during the revision of the manuscript.

1. L111: Authors should describe in detail how the test solutions were prepared.

2. L117-123: Detail parameters for both tungsten and glass probes should be reported, espacially the tip sizes.

3. L154-156: What is the rationale for this approach?

4. L170: Please add P value threshold for your tests.

5. L322-327: Please clearly address the "differences" the author mentioned.

6. L349-352: Are sucrose, glucose, nicotine, and tannin present in tobacco leaves? If present, what are the levels and how do they affect the results of the experiment?

7. L397-398: Why does 1 mmoL/L of nicotine elicit an electrophysiological response in the taste sensors, while 10 mmoL/L of nicotine has no effect on their feeding?

8. The practical application value of the findings should be discussed.

9. The discussion section is too long and trivial and needs to be consolidated and refined.

10. The aspect ratio of Figure 1 is incorrect, causing problems with the font presentation, please adjust.

11. It is better to find native English speaker to edit the M.S. 

Reviewer 2 Report

Comments and Suggestions for Authors

Round 2

Reviewer 2 Report

Comments and Suggestions for Authors
